# High rate of high-risk human papillomavirus among benign and breast cancer patients in Ethiopia

**Esmael Besufikad Belachew**[1,2,3]*, **Adey Feleke Desta**[2], **Andargachewu Mulu**[3], **Dinikisira Bekele Deneke**[4], **Dessalegn Abeje Tefera**[3], **Ashenafi Alemu**[3], **Endale Anberber**[5], **Daniel Beshah**[6], **Selfu Girma**[3], **Dareskedar Tsehay Sewasew**[3], **Tesfaye Sisay Tessema**[7], **Rawleigh Howe**[3]

1 Biology Department, College of Natural and Computational Sciences, Mizan Tepi University, Mizan, Ethiopia, 2 Department of Microbial, Cellular and Molecular Biology, College of Natural and Computational Sciences, Addis Ababa University, Addis Ababa, Ethiopia, 3 Armauer Hansen Research Institute, Addis Ababa, Ethiopia, 4 Department of Pathology, School of Medicine, Addis Ababa University, Addis Ababa, Ethiopia, 5 Department of Surgery, School of Medicine, Addis Ababa University, Addis Ababa, Ethiopia, 6 Department of Diagnostic Laboratory, Tikur Anbessa Specialized Hospital, College of Health Sciences, Addis Ababa University, Addis Ababa, Ethiopia, 7 Institute of Biotechnology, Addis Ababa University, Addis Ababa, Ethiopia

* getbb2006@gmail.com

**Data Availability Statement:** The data supporting the findings of this study cannot be shared publicly due to the absence of an agreement with patients for public raw data sharing. However, the data will

## Abstract

### Introduction

There have been numerous studies that showed the presence of human papillomavirus (HPV) in breast cancer; nonetheless, there is ongoing debate regarding their association. Given few studies in Ethiopia, we aimed to investigate the magnitude of HPV infection in Ethiopian breast cancer patients.

### Methods

A total of 120 formalin-fixed paraffin-embedded (FFPE) tissue blocks were obtained, and basic demographic, clinical, and histological data were collected from medical records. DNA was extracted from archived FFPE breast tissue specimens using GeneRead DNA FFPE Kit. The AnyplexTM II HPV28 Detection Kit (Seegene, Korea) was used to detect HPV by following the manufacturer's instructions. The SPSS Version 25 was used to enter and analyze data.

### Results

Among the 120 study participants; HPV (both high-risk and low-risk) was detected in 20.6% of breast cancer and 29.6% of non-malignant breast tumors. The most common genotype was the high-risk HPV 16 genotype. The frequency of HPV was nearly 10-fold higher in estrogen receptor-positive than ER-negative breast cancer. The percentage of HPV in the luminal (luminal A and luminal B) breast cancer subtypes was also much higher than in the non-luminal subtypes (HER-2 enriched and triple-negative breast cancer).

be made available to anyone requiring it for further research. Requesters will be required to ensure appropriate data storage. For access to the data supporting the study's conclusions, please contact the corresponding author, Esmael Besufikad Belachew (email: getbb2006@gmail.com), or the senior author, Rawleigh Howe (email: rawleigh. howe@ahri.gov.et). Although the authors cannot make their study's data publicly available at the time of publication, all authors commit to making the data underlying the findings described in this study fully available without restriction to those who request the data, in compliance with the PLOS Data Availability policy. For data sets involving personally identifiable information or other sensitive data, data sharing is contingent on the data being handled appropriately by the data requester and in accordance with all applicable local requirements.

**Funding:** This study was funded by Armauer Hansen Research Institute, Addis Ababa University, and Mizan Tepi University.

**Competing interests:** The authors declare that they have no competing interests.

## Conclusion

This study did not find a significant difference in HPV expression between breast cancer and non-malignant breast tumors; however, the higher percentage of HPV in ER-positive compared to ER-negative breast cancer warrants further attention.

## Introduction

The global burden of cancer has considerably increased in recent years, with an estimated 10 million people dying from cancer-related causes in 2020. Breast cancer is the most prevalent malignant tumor in women, accounting for 11.7% of all cancers. It is predicted that by 2040, breast cancer will result in more than 3 million new cases and 1 million deaths [1, 2]. While breast cancer in Africa comprises 8.3% of global cases, compared with 7.1% in North America and 6.4% in Western Europe, it contributes a disproportionate percentage of deaths (12.3%) [2]. Breast cancer is the most common type of cancer in Ethiopia, accounting for 31.9% of all female cancer cases in 2020, with 16,133 new cases and 9,061 deaths (27.5%) [1]. A considerable percentage of the world's burden of cancer may be linked to viruses, which are thought to be responsible for 15% of all human cancers globally. It has been demonstrated that human cancer can be caused by both DNA and RNA viruses [3]. Viruses such as human papillomavirus (HPV), epstein-barr virus (EBV), mouse mammary tumor virus (MMTV), and bovine leukemia virus (BLV) show potential roles in breast cancer development, but the epidemiological and proposed oncogenic mechanisms have not yet provided strong conclusive evidence [4, 5]. The main step in the development of malignant change is the integration of HPV infection into the genetic makeup of epithelial cells [6]. There is considerable debate and skepticism about the notion that HPV infection is the etiological factor contributing to the development of breast cancer [6]. The presence of HPV in malignant breast tumors has been reported, indicating a potential role of HPV in the early stage of breast cancer and breast carcinogenesis [7–11]. The existence of a causal link between HPV infection and breast cancer is still subject to debate; while some research revealed a higher risk consistent with a possible role of HPV in the development of breast cancer [5, 8, 10, 12–18], other studies found no such association [19–24]. Data, however, is limited in Ethiopia. Furthermore, breast and cervical cancer are the two most common types of cancer in Ethiopia. The primary cause of cervical cancer is HPV [25], which is also potentially associated with breast cancer. Hence, we aimed to investigate the magnitude of breast tissue-associated HPV infection among breast cancer patients in Ethiopia.

## Materials and methods

### Study participants

A total of 120 formalin-fixed paraffin-embedded (FFPE) tissue blocks were collected, 66 from breast cancer cases and 54 from cases with non-malignant breast tumors. Of the breast cancer cases, 23 were from Ayder Referral Hospital (Mekelle City, Tigray region), 9 from Hiwot Fana Specialized University Hospital (Harer City, Hareri Region), 9 from ALERT Specialized Hospital (Addis Ababa City), 13 from Jimma University Specialized Hospital (Jimma city, Oromia region), and 12 from Hawassa University Specialized Referral Hospital (Hawassa city, SNNP region). The benign breast tumors were enrolled from ALERT Specialized Hospital. Both breast cancer and non-malignant cases were selected based on the availability of clinical, pathological, and immunohistochemistry data. Cases that lacked this basic information were excluded from the study.

## Sample size

The sample size was calculated using Epi-info version 7.2.1.0, a statistical software. The characteristics that were taken into consideration were a 1:1 cases-to-control ratio, a 95% confidence level, 80% power, a 51.8% HPV prevalence among breast cancer cases, and an odds ratio of 3.0 [18]. Finally, a sample size of 130 was selected, of which 65 included cases and the remaining 65 controls. There were insufficient samples to collect the remaining 11 control cases, thus in the end, we used 66 cases, 54 controls, and 120 sample sizes together.

## Data collection

The demographic and histopathological data on females were collected from medical records. Age, sex, and place of residence were included, as well as clinicopathological data such as histologic type, stage, tumor size, grade, lymph node involvement, number of involved lymph nodes, and nature of the specimen. All the pathological data (histological type and grade) were obtained from histopathology slides prepared from collected tissue blocks in the Armauer Hansen Research Institute (AHRI). Polyclonal mouse anti-human estrogen receptor (ER) (DAKO clone Ep1; Agilent Technologies, Denmark) and anti-human progesterone receptor (PR) (DAKO clone PgR636, Agilent Technologies, Denmark) antibodies were used for staining. If the tumor had 1% or higher nuclear staining of tumor cells, it was considered ER/PR positive [26]. The polyclonal HER2/neu reagent (Agilent Technologies, Denmark) was used to perform the HER2/neu staining. HER2/neu negative specimens were defined as scores of 0 or 1+, while positive specimens were classified as scores of 3+, according to Wolff et al. (2018) recommendations for grading. Specimens having a score of 2 were considered equivocal [27]. A KI-67 percentage < 20% was categorized as a low proliferation index, whereas KI-67 ≥ 20% was classified as high proliferation according to the St. Gallen international panel of experts [28]. The immunohistochemistry results, which included ER, PR, and HER2 positivity, KI-67 percentage, and breast cancer subtype, were defined in a previous study [29].

## DNA extraction

DNA was extracted from stored (2015–2021) FFPE breast tissue specimens using GeneRead DNA FFPE Kit (Company, QIAGEN GmbH, QIAGEN Str. 1, D-40724 Hilden) following the manufacturer's protocol. DNA extraction was performed on ten tissue sections of 2 μm thickness in each sample. The 160 μL deparaffinization solution was then added and incubated for 3 minutes at 56°C. Following that, 55 μL of RNase-free water, 25 μL of fetch target buffer, and 20 μL of proteinase K were added. The mixture was then incubated for one hour at 56°C and another one hour at 90°C. The lower and clear phases were then transferred to a new microcentrifuge tube. The sample was then incubated in a thermomixer for 1 hour at 50°C with 115 μL RNase-free water and 35 μL uracil-N-glycosylase. Then 2 μL RNase A was added and incubated for 2 minutes at room temperature. The samples were then treated with 250 μL Buffer AL (lysis buffer) and 250 μL ethanol, and 700 μL lysate was applied to the QIAamp MinElute column. Each spin column was centrifuged after being filled with 500 μl Buffer AW1 (wash buffer 1). Then 500 μL Buffer AW2 (wash buffer 1) and 250 μL absolute ethanol were added. After adding 50 μL of Buffer ATE (elution buffer) to the middle of the membrane, it was left to incubate for five minutes at room temperature before centrifuging the mixture to extract the DNA. The quality of extracted DNA was checked using a Nanodrop 2000 spectrophotometer. All extracted DNA samples were then stored at -20°C until the PCR test was performed.

## Multiplex PCR

The PCR reactions were carried out using the CFX96 Deep well PCR instrument (Bio RAD, Singapore). AnyplexTM II HPV28 Detection Kit (Seegene, Korea) targeting the L1 gene was used to perform multiplex real-time PCR for the HPV genotyping as previously described [30]. It allows for the simultaneous amplification, detection, and separation of target nucleic acids from 19 high-risk HPV types (16, 18, 26, 31, 33, 35, 39, 45, 51, 52, 53, 56, 58, 59, 66, 68, 69, 73, 82) and 9 low-risk HPV types (6, 11, 40, 42, 43, 44, 54, 61, 70) as well as an internal control (IC). The human housekeeping gene (human beta-globin) was used as an endogenous IC to ensure extraction of DNA, verification of PCR reaction, and clarification of cell adequacy from each specimen. Both the high and low-risk HPV tests were performed using 5 μL of templates and 15 μL of PCR master mix with a total volume of 20 μL reaction mix. For negative control (NC), we used 5 μL of RNase-free water instead of template nucleic acid, and for positive control (PC), used 5 μL of each HPV28 PC1, PC2, and PC3. Each PC includes clones for 5 targets in A set (14 types of high risk and IC) and 5 targets in B set (5 types of high risk, 9 types of low risk and IC).

## Ethical approval and consent

Ethical approval for this study was obtained from the College of Natural Science Institutional Ethics Review Board (CNS-IRB) Addis Ababa University (No. IRB/032/2018), AHRI/ALERT Ethics Review Committee (AAERC) (No. PO/27/19) and Federal Democratic Republic of Ethiopia Ministry of Education (No.7/2.12/m259/35). The author accessed the data on September 21, 2022; however, the lead investigator is the only one with access to patient information.

## Statistical analysis

Data from the PCR result, the pathology report, and the IHC results were entered and analyzed using SPSS Version-25 software. Univariate Chi-square tests were used to assess the association between the predictor and the outcome variables. Logistic regression was used to identify the association between a particular predictor and outcome variables after adjusting for the effects of all other predictors. For statistical significance, a p-value of less than 0.05 was used.

# Results

## Magnitude of HPV infection and genotypes

A total of 120 study participants were enrolled in this study, of which 30 (25.0%) were positive for both high-risk and low-risk HPV infection. The frequency of high-risk and low-risk HPV was found in 20.6% of breast cancer cases, but it was found in 29.6% of non-malignant breast tumor cases. The high-risk HPV 16 genotype was dominant with 76.6% and 68.7% accounting for breast cancer cases and benign tumors, respectively (Table 1).

## Associated factors

The most common histologic type of breast cancer (75.8%) was invasive carcinoma of no special type, followed by invasive lobular carcinoma (6.1%). Among all cases aged < 50 years, HPV-positive cases represented 26.5%, whereas among those aged > 50, a smaller percentage, 16.7% were positive for HPV; however, these differences were not statistically significant (p = 0.38), consistent with previous studies in Ethiopia [29, 31].

Estrogen receptor-positive breast cancer had a significantly higher proportion of HPV infection than estrogen receptor-negative breast cancer (p = 0.022. The luminal IHC subtypes (luminal A and Luminal B) together had a significantly higher proportion of HPV than the

**Table 1. The magnitude of HPV infection and genotypes among breast cancer and non-malignant breast cases.**

| Characteristics | | | Case (N = 66) | Control (N = 54) | Total (N = 120) |
|---|---|---|---|---|---|
| HPV infection | HPV positive | | 14(20.6%) | 16(29.6%) | 30(25.0%) |
| | HPV negative | | 52(76.4%) | 38(70.4%) | 90(75.0%) |
| | Total | | 66(100%) | 54(100%) | 120(100%) |
| HPV genotype | | | Case (N = 14) | Control (N = 16) | Total (N = 30) |
| | High Risk HPV | HPV 16 | 12(85.6%) | 11(68.7%) | 23(76.6%) |
| | | HPV 59 | 1(7.2% | 0 | 1(3.3%) |
| | | HPV 31 | 0 | 1(6.3%) | 1(3.3%) |
| | Low Risk HPV | HPV 44 | 1(7.2%) | 0 | 1(3.3%) |
| | | HPV 6 | 0 | 1(6.3%) | 1(3.3%) |
| | | HPV 42 | 0 | 1(6.3%) | 1(3.3%) |
| | Mixed (High & low Risk HPV) | | | | |
| | | HPV 16 and 42 | 0 | 1(6.3%) | 1(3.3%) |
| | | HPV 31 and 45 | 0 | 1(6.3%) | 1(3.3%) |
| | | Total | 14(100%) | 16(100%) | 30(100%) |

non-luminal subtypes (HER-2 enriched and triple-negative) (p = 0.018). The percentage of HPV was also higher among HER-2 grades 0 than among other grades; however, this was not statistically significant. Cases with a low KI-67 proliferation index had a higher HPV percentage than those with a high KI-67 index, but this only reached borderline significance (p = 0.056). There was no significant association between tumor size or lymph node involvement and HPV infection (Table 2).

In multivariate logistic regression analysis (Table 3), a total of 58 breast cancer cases were analyzed, and the ER-positive breast cancer cases were almost 10 times more likely to have HPV than ER-negative breast cancer cases, and this retained statistical significance (p = 0.043) after correction for other variables. Cases with high KI-67 proliferation index were not statistically significantly different in the adjusted multivariate model (**Table 3**).

## Discussion

The present study evaluated the association of HPV infection, using molecular PCR-based assays of formalin-fixed paraffin-embedded tissue, of breast cancer cases in comparison with non-malignant breast tumors. We did not observe a statistically significant difference in the percentage of either high or low-risk HPV in malignant versus non-malignant breast tumors. However, further analysis of the breast cancer cases revealed an almost 10-fold difference in HPV percentage in ER-positive as opposed to ER-negative tissues. To our knowledge, this represents the first study to show the association of HPV with ER expression in breast cancer tissue.

There is no concrete evidence to support the oncogenic role of HPV in breast cancer [32]. However, persistent HPV infection causes the host genome to integrate with the viral genome, increasing genomic instability and the inactivation of cell cycle checkpoints, which ultimately leads to the development of cancer [32]. The controversial role of HPV in the development of breast cancer is documented [33].

In this study, the percentage of HPV (both high-risk and low-risk) was 25.0%. Non-malignant breast tumors had a higher HPV percentage (29.6%) than breast cancer (20.6%), which may suggest that there is no association between HPV infection and breast cancer. Another earlier study found that HPV was present in 2.7% of Ethiopian breast cancer cases [34] which is a far lower number than the current study. The percentage of HPV positivity was higher in

**Table 2. Associated factors for HPV infection among breast cancer cases.**

| Characteristics | | HPV infection | | Total | P-value* |
|---|---|---|---|---|---|
| | | Positive | Negative | | |
| Laterality | Right | 14(46.7%) | 46(51.1%) | 60(50.0%) | 0.673 |
| | Left | 16(53.3%) | 44(48.9%) | 60(50.0%) | |
| | Total | 30(100%) | 90(100.0%) | 120(100%) | |
| Tumor Size | T1 & T2 | 3(30.0%) | 19(44.2%) | 22(41.5%) | 0.412 |
| | T3 & T4 | 5(70.0%) | 24(55.8%) | 31(58.5%) | |
| | Total | 10(100%) | 43(100%) | 53(100%) | |
| Lymph node involvement | Yes | 7(50.0%) | 26(55.3%) | 33(54.1%) | 0.726 |
| | No | 7(50.0%) | 21(44.7%) | 28(45.9%) | |
| | Total | 14(100%) | 47(100%) | 61(100%) | |
| Stage | Stage I | 0 (0%) | 1(2.3%) | 1(1.8%) | 0.682 |
| | Stage II | 5 (50%) | 20(46.5%) | 25(47.2%) | |
| | Stage III | 5 (50%) | 22(51.2%) | 27(51.0%) | |
| | Total | 10 (100%) | 43 (100%) | 53(100%) | |
| Grade | I | 5(35.7%) | 22(42.3%) | 27(40.9%) | 0.824 |
| | II | 4(28.6%) | 11(21.2%) | 15(22.7%) | |
| | III | 5(35.7%) | 19(36.5%) | 24(36.4%) | |
| | Total | 14(100%) | 52(100%) | 66(100%) | |
| Histological type | Invasive carcinoma of no special type | 12(85.8%) | 38(73.1%) | 50(75.8%) | 0.789 |
| | Invasive lobular carcinoma | 1(7.1%) | 3(5.8%) | 4(6.1%) | |
| | Mucinous carcinoma | 0(0%) | 2(3.8%) | 2(3.0%) | |
| | Metaplastic carcinoma | 0(0%) | 2(3.8%) | 2(3.0%) | |
| | Others | 1(7.1%) | 7(13.5%) | 8(12.1%) | |
| | Total | 14(100%) | 52(100%) | 66(100%) | |
| ER | Positive | 12(85.7%) | 27(51.9%) | 39(59.1%) | 0.022 |
| | Negative | 2(14.3) | 25(48.2%) | 27(40.9%) | |
| | Total | 14(100%) | 52(100%) | 66(100%) | |
| PR | Positive | 7(50.0%) | 24(46.2%) | 31(47.0%) | 0.798 |
| | Negative | 7(50.0%) | 28(53.8%) | 35(53.0%) | |
| | Total | 14(100%) | 52(100%) | 66(100%) | |
| HER-2 | Positive | 2(14.3%) | 14(26.9%) | 16(24.2%) | 0.617 |
| | Negative | 10(71.4%) | 32(61.5%) | 42(63.6%) | |
| | Equivocal | 2(14.3%) | 6(11.6%) | 8(12.2%) | |
| | Total | 14(100%) | 52(100%) | 66(100%) | |
| HER-2 Grade | IHC 0 | 9(64.3%) | 23(44.2%) | 32(48.5%) | 0.400 |
| | IHC1 | 2(14.3%) | 9(17.3%) | 11(16.7%) | |
| | IHC 2 | 2(14.3%) | 6(11.5%) | 8(12.1%) | |
| | IHC3 | 1(7.1%) | 14(26.9%) | 15(22.7%) | |
| | Total | 14(100%) | 52(100%) | 66(100%) | |
| KI-67 | Low proliferation | 11(78.6%) | 26(50.0%) | 37(56.1%) | 0.056 |
| | High proliferation | 3(21.4%) | 26(50.0%) | 29(43.9%) | |
| | Total | 14(100%) | 52(100%) | 66(100%) | |
| Subtype | Luminal | 11(91.7%) | 25(55.6%) | 36(62.1%) | 0.018 |
| | Non-luminal | 1(8.3%) | 21(44.4%) | 22(37.9%) | |
| | Total | 12(100%) | 45(100%) | 58(100%) | |

*Differences of features among HPV infection assessed by $X^2$ test.

**Table 3. Multivariate logistic regression of ER, KI-67 proliferation index, and HER-2 status, taken as independent variables for HPV expression dependent variables.**

| Independent variables | | HPV positive | HPV negative | Total† | COR (95% CI) * | p-value | AOR (95% CI) | p-value |
|---|---|---|---|---|---|---|---|---|
| ER | Positive | 12(85.7%) | 27(51.9%) | 39(59.1%) | 5.56(1.1–27.3) | 0.035 | 9.61(1.1–86.3) | 0.043 |
| | Negative Ref. | 2(14.3) | 25(48.2%) | 27(40.9%) | | | | |
| HER-2 | Positive Ref. | 2(16.7%) | 14(30.4%) | 16(27.6%) | 2.19(0.4–11.3) | 0.350 | 1.01(0.2–6.5) | 0.989 |
| | Negative | 10(83.3%) | 32(69.6%) | 42(72.4%) | | | | |
| KI-67 | Low proliferation | 11(78.6%) | 26(50.0%) | 37(56.1%) | 3.67(0.9–14.7) | 0.066 | 2.29(0.5–10.7) | 0.291 |
| | High proliferation Ref. | 3(21.4%) | 26(50.0%) | 29(43.9%) | | | | |

*COR -crude odds ratio, AOR-adjusted odds ratio

† HER-2 equivocal cases were not included

the current study than in Sudan (8.6%) and Congo (15%) [35]. The difference could be explained by the sensitivity of the PCR assay employed or other HPV-associated risk factors. The common genotype in this study was HPV-16, also reported in other studies [35, 36]. The specific role played by this virus in the pathogenesis of breast cancer needs careful consideration through larger epidemiologic studies.

A novel finding in our study was that ER-positive breast cancer exhibited an HPV percentage almost 10 times greater than ER-negative breast cancer. A connection between the HPV virus and ER-positive breast cancer was also reported by other investigations [37, 38]. A few studies have shown that ER expression may encourage cervical neoplasia by altering the expression of the HPV genes [39, 40]. Estrogen exposure causes DNA double-strand breaks that ultimately result in genomic instability and carcinogenesis via G protein-coupled receptor 30 for cervical cancer and cervical dysplasia, however, the exact mechanism is yet unknown [41–44]. These arguments might also apply to breast cancer development, but further molecular studies are needed to determine if and how HPV infections are associated with ER-positive breast cancer.

Previous studies have explored possible mechanisms in which HPV could be associated with HER-2 and Ki67 levels. In vitro studies with breast cancer cell lines indicated that transfected HPV E6 and E7 could enhance HER-2 expression and confer an EGF-independent in vitro proliferation, suggesting a possible mechanism for how HPV infection might contribute to breast cancer progression [45, 46]. KI-67 expression is related to the proliferation index of breast cancer cases. A study observed that breast cancer with low proliferation indices had high HPV prevalence [46]. Similarly, a low histopathological grade of vaginal carcinoma is associated with high HPV prevalence [47]. In our study, we observed differences between HER-2 expression and KI-67 proliferation indices among HPV-positive and negative breast cancer cases; however, these differences were not statistically significant, particularly after adjustment in a multivariate model.

## Conclusion

This study did not find a significant difference in HPV infection prevalence between breast cancer and benign breast tumors. We did observe, for the first time, that HPV prevalence was much higher in ER-positive breast cancer than in ER-negative breast cancer, and this observation warrants further attention.

## Acknowledgments

The authors acknowledged all staff of the pathology and molecular department and driver officers (Mr. Shambel and Mr. Solomon Gebre) and drivers from Armauer Hansen Research Institute for their contributions.

## Author Contributions

**Conceptualization:** Esmael Besufikad Belachew, Adey Feleke Desta, Andargachewu Mulu, Dinikisira Bekele Deneke, Dareskedar Tsehay Sewasew, Tesfaye Sisay Tessema, Rawleigh Howe.

**Data curation:** Esmael Besufikad Belachew, Adey Feleke Desta, Endale Anberber, Selfu Girma, Tesfaye Sisay Tessema, Rawleigh Howe.

**Formal analysis:** Esmael Besufikad Belachew, Tesfaye Sisay Tessema, Rawleigh Howe.

**Investigation:** Esmael Besufikad Belachew, Rawleigh Howe.

**Methodology:** Esmael Besufikad Belachew, Dessalegn Abeje Tefera, Ashenafi Alemu, Daniel Beshah, Selfu Girma, Rawleigh Howe.

**Resources:** Rawleigh Howe.

**Supervision:** Adey Feleke Desta, Andargachewu Mulu, Dinikisira Bekele Deneke, Tesfaye Sisay Tessema, Rawleigh Howe.

**Validation:** Esmael Besufikad Belachew, Dessalegn Abeje Tefera, Ashenafi Alemu.

**Writing – original draft:** Esmael Besufikad Belachew, Andargachewu Mulu, Dinikisira Bekele Deneke, Dessalegn Abeje Tefera, Ashenafi Alemu, Endale Anberber, Daniel Beshah, Selfu Girma, Dareskedar Tsehay Sewasew, Tesfaye Sisay Tessema, Rawleigh Howe.

**Writing – review & editing:** Esmael Besufikad Belachew, Adey Feleke Desta, Andargachewu Mulu, Dinikisira Bekele Deneke, Dessalegn Abeje Tefera, Ashenafi Alemu, Endale Anberber, Daniel Beshah, Selfu Girma, Dareskedar Tsehay Sewasew, Tesfaye Sisay Tessema, Rawleigh Howe.

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
