## [Decision Letter · Decision Letter 0]

2 Jan 2024

PONE-D-23-38019High rate of high-risk HPV among benign and breast cancer patients in EthiopiaPLOS ONE

Dear Dr. Belachew,

Thank you for submitting your manuscript to PLOS ONE. After careful consideration, we feel that it has merit but does not fully meet PLOS ONE’s publication criteria as it currently stands. Therefore, we invite you to submit a revised version of the manuscript that addresses the points raised during the review process.

We look forward to receiving your revised manuscript.

Kind regards,

Kazunori Nagasaka

Academic Editor

PLOS ONE

https://bmcresnotes.biomedcentral.com/articles/10.1186/s13104-023-06518-5

In your revision ensure you cite all your sources (including your own works), and quote or rephrase any duplicated text outside the methods section. Further consideration is dependent on these concerns being addressed.

A clean copy of the edited manuscript (uploaded as the new *manuscript* file)”.

Additional Editor Comments:

Dear Authors,

The manuscript is very intriguing, but some revision is needed.

Please revise the manuscript accoring to the reviewer's comments.

Sincerely,

Plos One editorial office

Reviewers' comments:

Reviewer's Responses to Questions

**Comments to the Author**

1. Is the manuscript technically sound, and do the data support the conclusions?

Reviewer #1: No

Reviewer #2: Partly

Reviewer #3: Yes

Reviewer #4: Partly

2. Has the statistical analysis been performed appropriately and rigorously? 

Reviewer #1: No

Reviewer #2: Yes

Reviewer #3: No

Reviewer #4: Yes

3. Have the authors made all data underlying the findings in their manuscript fully available?

Reviewer #1: No

Reviewer #2: Yes

Reviewer #3: Yes

Reviewer #4: Yes

4. Is the manuscript presented in an intelligible fashion and written in standard English?

Reviewer #1: No

Reviewer #2: Yes

Reviewer #3: No

Reviewer #4: No

5. Review Comments to the Author

Reviewer #1: 1-The Introduction was well described.

2-The Materials and methods. The study was not well-designed. Since the results, is controversial. The authors have reported 14/66(20.6%) cases were positive for HPV while 16/54(29.6%) control were positive for HPV high risk groups or low risk groups? It seems some problems in samples collection or some technical problems have been taken place in malignancy tumor and even among the control case (benign) detections reports by pathologist.

3- Such work it requires detection and expression of HPV 16 E6 mRNA, HPV 31 E6 mRNA, HPV 42 E6 mRNA, HPV 44 E6 mRNA, HPV 45 E6 mRNA, HPV 59 E6 mRNA, and HPV 16 E7 mRNA, HPV 31 E7 mRNA, HPV 42 E7 mRNA, HPV 44 E7 mRNA, HPV 45 E7 mRNA, HPV 59 E7 mRNA among the malignancy and beings tumors’ samples.

4- Therefore, the results and discussion cannot be justified.

5-No novelty.

Reviewer #2: This manuscript tackles a controversial area of research that has not conclusively established a causal link between breast cancer and HPV infection. The authors investigated a well -established fact that HPV infection occurs in breast cancer tissue. They examined the extent of HPV infection in Ethiopian cases -though whether female or male cases were chosen is not described. Their novel finding is the differentially high distribution of HPV infection in ER+ tumours

Would the authors please address the following queries

INTRODUCTION

1) In the introduction please provide your rationale for selecting HPV over the other viruses mentioned for exploration in your Ethiopian sample

2) I suggest that from the discussion section, lines 191 starting from The existence of a causal link....202 should be put into the introduction section as part of the rationale for studying HPV and its potential aetiological link to breast cancer development.

METHODS SECTION

1) Please explain your rationale for sample selection and there is no sample size calculation to justify your sample size selection -how can you be sure that your sample size is sufficient to give meaningful results- particularly with respect to distribution of receptor subtyping -is this representative of population receptor subtype distributions. In addition your regression analysis is not described in the analysis section of your methods -please correct this.

a) 85% of the sample selected were from very young patients- is this the typical age distribution of breast cancer among the Ethiopian population?

b) Stage at diagnosis is not presented in table 2 and should be given to give an accurate representation of size, nodal involvement and metastatic invasion- what proportion of patients had stage 3 &4 disease?

2) Table 2 confuses me - your heading says associated factors for HPV infection among BC and non-malignant breast cases -yet tumour size, lymph nodes grade receptor subtyping is only for cases- I feel it would be much clearer if you had major column headings as CASES and CONTROLS and TOTALS with subheadings under each as HPV+ HPV-. Naturally describing the tumour features would be left blank for controls -but as it stands your table is confusing- also HER2 should be expressed as negative, equivocal and positive.

3) For Table 3 you have not defined COR and AOR -is COR crude odds ratio and AOR adjusted odds ration -if so then include these definitions in your footer

DISCUSSION SECTION

1)Please first summarise your major findings in a first paragraph

2) Move lines 191-202 to introduction section as part of rationale for selecting HPV from the other viruses to study

3) Line 212 add the word "development"

4) line 214 clarify what you mrean by "long-term breast cancer" -is this advanced breast cancer?

Reviewer #3: The article explores an intriguing topic.

Strengthening the writing style could potentially elevate the quality of the work, providing a more polished and engaging reading experience for the audience

Please improve the whole manuscript writing in a way of methodology, table resut display, analysis and discussion.

Reviewer #4: Congratulations to the authors for undertaking this research project which aims to add to the body of knowledge concerning Breast cancer & HPV, which remains one of the latest controversial issues. Some concerns I have with the manuscripts:

1. The aim of the study was to investigate the magnitude of HPV infection among breast cancer in Ethiopia, however, the conclusion speaks of ' ....no proof of a link between HPV and breast cancer'-How did they determine that there was no link or association? How did they get to this conclusion? (Sentence 181 & conclusion 224)

2. Sentence 190-191: As above, the author cannot make the implication that implication solely on the fact that HPV % was higher in benign than cancerous lesions. Moreover, instead of 'imply' perhaps the author may write ' which may suggest'

3. Sentence 184: Needs paraphrasing

4. Sentence 185: No reference

5. Sentence 188:.......'documented' instead of 'indicated'

Sentence 201-202: '...does not provide evidence to support HPV's involvement in the development & progression of breast cancer'- This was not determined in this study.

6. PLOS authors have the option to publish the peer review history of their article (what does this mean?). If published, this will include your full peer review and any attached files.

Reviewer #1: No

Reviewer #2: **Yes: **Dr Maureen Joffe

Reviewer #3: No

Reviewer #4: **Yes: **Dr Boitumelo Phakathi

---

## [Author Response · Author response to Decision Letter 0]

16 Jan 2024

The comments of editors and reviewers' comments were constructive and valuable in enhancing the manuscript. We thank all the editors and reviewers for their crucial comments and suggestions. We reflected on a few comments here and all the necessary changes were incorporated into the manuscript in a clean version and with track changes.

Reviewer #1: 

1-The Introduction was well described. 

Response: Thank you for your supportive comments

2-The Materials and methods. The study was not well-designed. Since the results are controversial. 

The authors have reported that 14/66(20.6%) cases were positive for HPV while 16/54(29.6%) control was positive for HPV high-risk groups or low-risk groups. 

Response: Thank you for your comment, this result is for both high-risk and low-risk HPV, and we have included it in the updated version (lines 35, ,306)

It seems some problems in sample collection or some technical problems have taken place in malignancy tumors and even among the control case (benign) detections reports by pathologists.

Response: Our pathologists examined all of the slides and made classification according to standard diagnostic procedures to determine whether the breast tumor was benign or malignant (They used Robbins Basic Pathology as a reference). The benign breast tumors included in this study were fibroadenoma and fibrocystic changes.

3- Such work requires detection and expression of HPV 16 E6 mRNA, HPV 31 E6 mRNA, HPV 42 E6 mRNA, HPV 44 E6 mRNA, HPV 45 E6 mRNA, HPV 59 E6 mRNA, and HPV 16 E7 mRNA, HPV 31 E7 mRNA, HPV 42 E7 mRNA, HPV 44 E7 mRNA, HPV 45 E7 mRNA, HPV 59 E7 mRNA among the malignancy and beings tumors’ samples.

4- Therefore, the results and discussion cannot be justified.

5-No novelty.

Response: Thank you, this feedback is valuable. In this study, to our knowledge, we found an association with HPV and ER for the first time (Line 518, 600). We understand detection and expression of E6 mRNA and E7 mRNA among HPV-positive cases is important, and we will consider it for our future study. 

Reviewer #2: This manuscript tackles a controversial area of research that has not conclusively established a causal link between breast cancer and HPV infection. The authors investigated a well-established fact that HPV infection occurs in breast cancer tissue. They examined the extent of HPV infection in Ethiopian cases -though whether female or male cases were chosen is not described. Their novel finding is the differentially high distribution of HPV infection in ER+ tumors-

Response: Thank you for your encouraging comments, we have only chosen females and are now included in the method section line 175.

Would the authors please address the following queries?

INTRODUCTION

1) In the introduction please provide your rationale for selecting HPV over the other viruses mentioned for exploration in your Ethiopian sample- 2) I suggest that from the discussion section, lines 191 starting from The existence of a causal link....202 should be put into the introduction section as part of the rationale for studying HPV and its potential etiological link to breast cancer development. 

Response: Thank you for your comments, we added and moved your suggested discussion section into the introduction as part of the rationale, lines 84-87. Another rationale is also indicated in line 87, 124,125. 

METHODS SECTION

1) Please explain your rationale for sample selection there is no sample size calculation to justify your sample size selection -how can you be sure that your sample size is sufficient to give meaningful results- particularly for the distribution of receptor subtyping -is this representative of population receptor subtype distributions? 

Response: Your comments are appreciated; this study was a pilot exploratory. We have calculated the sample size using Epi-info version 7.2.1.0, a statistical software indicated in lines 141-147 

In addition, your regression analysis is not described in the analysis section of your methods -please correct this. 

Response: Your comments are appreciated, based on your comments lines 293 to 296 have been updated.

a) 85% of the sample selected were from very young patients- is this the typical age distribution of breast cancer among the Ethiopian population?

Response: Thank you for your comment, Ethiopian breast cancer patients fell within the younger age group category in general(indicated in lines 315, 316, 335 and 336

b) Stage at diagnosis is not presented in Table 2 and should be given to give an accurate representation of size, nodal involvement, and metastatic invasion- what proportion of patients had stage 3 &4 disease?

Response: Thank you for your comment, based on your comment, Table 2 has been updated.

2) Table 2 confuses me - your heading says associated factors for HPV infection among BC and non-malignant breast cases -yet tumor size, lymph nodes grade receptor subtyping is only for cases- I feel it would be much clearer if you had major column headings as CASES and CONTROLS and TOTALS with subheadings under each as HPV+ HPV-. Naturally describing the tumor features would be left blank for controls -but as it stands your table is confusing- also HER2 should be expressed as negative, equivocal, and positive.

3) For Table 3 you have not defined COR and AOR -is COR crude odds ratio and AOR adjusted odds ratio -if so then include these definitions in your footer

Response: Thank you for your crucial comments; Table 2 was unclear. To avoid this confusion, we choose to write the age group data in text form. As a result, Table 2 now only includes breast cancer cases. We included the HER-2 equivocal cases in the chi-square test but excluded them from the regression analysis because we only used binary independent variables.

In the footer of Table 3, we have also included definitions for the COR crude odds ratio and AOR adjusted odds ratio.

DISCUSSION SECTION

1)Please first summarise your major findings in the first paragraph

Response: Thank you for your comment, as per your comments, included our major finding in the first paragraph of the discussion.

2) Move lines 191-202 to the introduction section as part of the rationale for selecting HPV from the other viruses to study

Response: Thank you for your comment, as per your comments, we moved it to the introduction line 84-87.

3) Line 212 adds the word "development"

Response: Thank you for your comments, we have included them in line 586.

4) line 214 clarifies what you mean by "long-term breast cancer" -is this advanced breast cancer?--- 

Response: To avoid reader confusion we have removed this term indicated in line 589.

Reviewer #3: The article explores an intriguing topic. Strengthening the writing style could potentially elevate the quality of the work, providing a more polished and engaging reading experience for the audience- 

Please improve the whole manuscript writing in a way of methodology, table result display, analysis, and discussion. 

Response: Thank you for your encouraging comments, we improve and strengthen the writing style.

Reviewer #4: Congratulations to the authors for undertaking this research project which aims to add to the body of knowledge concerning Breast cancer & HPV, which remains one of the latest controversial issues. Some concerns I have with the manuscripts:

1. The study aimed to investigate the magnitude of HPV infection among breast cancer in Ethiopia, however, the conclusion speaks of ' ....no proof of a link between HPV and cancer '. ' How, did they determine that there was no link or association? How did they get to this conclusion? (Sentence 181 & conclusion 224)

Response: Comment accepted and included, and this sentence is now modified 518 and 532 and throughout the manuscript. 

2. Sentence 190-191: As above, the author cannot make the implication that implication solely on the fact that HPV % was higher in benign than cancerous lesions. Moreover, instead of 'imply' perhaps the author may write ' which may suggest'

Response: Thank you for your comment, the comment is accepted and included in line 540

3. Sentence 184: Needs paraphrasing

Response: Thank you for your comment, done, line 534

4. Sentence 185: No reference- 

Response: This is the reference used for “The role of integration in oncogenic progression of HPV-associated cancers” for this sentence, which is similar to the below sentence. But to avoid confusion we have added the same reference in both sentences.

5. Sentence 188:.......' documented' instead of 'indicated'-

Response: Comment accepted and included in line 538

Sentence 201-202: '...does not provide evidence to support HPV's involvement in the development & progression of breast cancer'- This was not determined in this study.

 Response: Thank you for your comment, the comment is accepted and included and this sentence is now modified in the whole manuscript

---

## [Editor Report · Decision Letter 1]

22 Jan 2024

PONE-D-23-38019R1High rate of high-risk human papillomavirus among benign and breast cancer patients in EthiopiaPLOS ONE

Dear Dr. Belachew,

Thank you for submitting your manuscript to PLOS ONE. After careful consideration, we feel that it has merit but does not fully meet PLOS ONE’s publication criteria as it currently stands. Therefore, we invite you to submit a revised version of the manuscript that addresses the points raised during the review process.

We look forward to receiving your revised manuscript.

Kind regards,

Kazunori Nagasaka

Academic Editor

PLOS ONE

Journal Requirements:

**Additional Editor Comments:**

Dear Authors,

Thank you for your submission to Plos One.

I think most of the concerns are explained and fully revised, but I have a little concern that HPV infections are both high risk and low risk types.

Could you divide this population into high and low risk HPV type groups and look at the differences?

Sincerely,

Kazunori Nagasaka

---

## [Author Response · Author response to Decision Letter 1]

23 Jan 2024

The comments of both editors were constructive and valuable in enhancing the manuscript. We thank all the editors for their crucial comments and suggestions. We reflected on a few comments here and all the necessary changes were incorporated into the manuscript in a clean version and with track changes.

Academic Editor: Please review your reference list to ensure that it is complete and correct. If you have cited papers that have been retracted, please include the rationale for doing so in the manuscript text, or remove these references and replace them with relevant current references. Any changes to the reference list should be mentioned in the rebuttal letter that accompanies your revised manuscript. If you need to cite a retracted article, indicate the article’s retracted status in the References list and also include a citation and full reference for the retraction notice.

Response: all references have been updated. However, we have removed reference 41 (OHBA, K., CHONG, P. P. & YAMAMOTO, N. 2017. Role of Human Papillomavirus (HPV), Estrogen and Apobec3B Axis in Breast Cancer Initiation Line 216) as it has now been retracted.

 Additional Editor Comments:

Dear Authors,

Thank you for your submission to Plos One.

I think most of the concerns are explained and fully revised, but I have a little concern that HPV infections are both high-risk and low-risk types.

Could you divide this population into high and low-risk HPV-type groups and look at the differences?

Response: This feedback is really helpful, and we have amended it in Table 1 . Line 156. Because there are fewer than ten cases in the low-risk HPV groups, regression, and chi-square analysis cannot be performed to further examine it. In most circumstances, group variables of less than 10 should not be compared for statistical regression and chi-square analysis.

---

## [Editor Report · Decision Letter 2]

29 Jan 2024

High rate of high-risk human papillomavirus among benign and breast cancer patients in Ethiopia

PONE-D-23-38019R2

Dear Dr. Belachew,

We’re pleased to inform you that your manuscript has been judged scientifically suitable for publication and will be formally accepted for publication once it meets all outstanding technical requirements.

Kind regards,

Kazunori Nagasaka

Academic Editor

PLOS ONE

Additional Editor Comments (optional):

Dear authors,

I think the manuscript is much improved and acceptable in Plos One.

Thank you for your submission.

Yours sincerely,

Plos One
---

## [Editor Report · Acceptance letter]

12 Mar 2024

PONE-D-23-38019R2 

PLOS ONE

Dear Dr. Belachew, 

I'm pleased to inform you that your manuscript has been deemed suitable for publication in PLOS ONE. Congratulations! Your manuscript is now being handed over to our production team.

Kind regards, 

on behalf of

Professor Kazunori Nagasaka 

Academic Editor

PLOS ONE